# Hypothesis Transfer Learning via Transformation Functions

**Simon S. Du**
Carnegie Mellon University
ssdu@cs.cmu.edu

**Jayanth Koushik**
Carnegie Mellon University
jayanthkoushik@cmu.edu

**Aarti Singh**
Carnegie Mellon University
aartisingh@cmu.edu

**Barnabás Póczos**
Carnegie Mellon University
bapoczos@cs.cmu.edu

## Abstract

We consider the Hypothesis Transfer Learning (HTL) problem where one incorporates a hypothesis trained on the source domain into the learning procedure of the target domain. Existing theoretical analysis either only studies specific algorithms or only presents upper bounds on the generalization error but not on the excess risk. In this paper, we propose a unified algorithm-dependent framework for HTL through a novel notion of *transformation function*, which characterizes the relation between the source and the target domains. We conduct a general risk analysis of this framework and in particular, we show for the first time, if two domains are related, HTL enjoys faster convergence rates of excess risks for Kernel Smoothing and Kernel Ridge Regression than those of the classical non-transfer learning settings. Experiments on real world data demonstrate the effectiveness of our framework.

## 1 Introduction

In a classical transfer learning setting, we have a large amount of data from a source domain and a relatively small amount of data from a target domain. These two domains are related but not necessarily identical, and the usual assumption is that the hypothesis learned from the source domain is useful in the learning task of the target domain.

In this paper, we focus on the regression problem where the functions we want to estimate of the source and the target domains are different but related. Figure 1a shows a 1D toy example of this setting, where the source function is $f^{so}(x) = \sin(4\pi x)$ and the target function is $f^{ta}(x) = \sin(4\pi x) + 4\pi x$. Many real world problems can be formulated as transfer learning problems. For example, in the task of predicting the reaction time of an individual from his/her fMRI images, we have about 30 subjects but each subject has only about 100 data points. To learn the mapping from neural images to the reaction time of a specific subject, we can treat all but this subject as the source domain, and this subject as the target domain. In Section 6, we show how our proposed method helps us learn this mapping more accurately.

This paradigm, hypothesis transfer learning (HTL) has been explored empirically with success in many applications [Fei-Fei et al., 2006, Yang et al., 2007, Orabona et al., 2009, Tommasi et al., 2010, Kuzborskij et al., 2013, Wang and Schneider, 2014]. Kuzborskij and Orabona [2013, 2016] pioneered the theoretical analysis of HTL for linear regression and recently Wang and Schneider [2015] analyzed Kernel Ridge Regression. However, most existing works only provide generalization bounds, i.e. the difference between the true risk and the training error or the leave-one-out error. These analyses are not complete because minimizing the generalization error does not necessarily

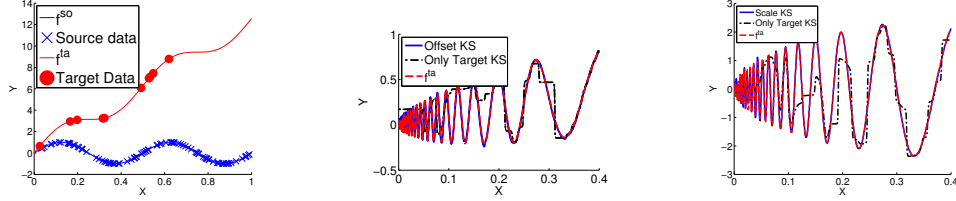

(a) A toy example of transfer learning. We have many more samples from the source domain than the target domain.

(b) Transfer learning with Offset Transformation.

(c) Transfer learning with Scale Transformation.

Figure 1: Experimental results on synthetic data.

reduce the true risk. Further, these works often rely on a particular form of transformation from the source domain to the target domain. For example, Wang and Schneider [2015] studied the offset transformation that instead of estimating the target domain function directly, they learn the residual between the target domain function and the source domain function. It is natural to ask what if we use other transfer functions and how it affects the risk on the target domain.

In this paper, we propose a general framework of HTL. Instead of analyzing a specific form of transfer, we treat it as an input of our learning algorithm. We call this input *transformation function* since intuitively, it captures the relevance between these two domains.[1] This framework unifies many previous works Wang and Schneider [2014], Kuzborskij and Orabona [2013], Wang et al. [2016] and naturally induces a class of new learning procedures.

Theoretically, we develop excess risk analysis for this framework. The performance depends on the stability [Bousquet and Elisseeff, 2002] of the algorithm used as a subroutine that if the algorithm is stable then the estimation error in the source domain will not affect the estimation in the target domain much. To our knowledge, this connection was first established by Kuzborskij et al. [2013] in the linear regression setting but here we generalize it to a broader context. In particular, we provide explicit risk bounds for two widely used nonlinear estimators, Kernel Smoothing (KS) estimators and Kernel Ridge Regression (KRR) as subroutines. To the best of our knowledge, these are the first results showing when two domains are related, transfer learning techniques have faster statistical convergence rate of excess risk than that of non-transfer learning of kernel based methods. Further, we accompany this framework with a theoretical analysis showing a small amount of data for cross-validation enables us (1) avoid using HTL when it is not useful and (2) choose the best transformation function as input from a large pool.

The rest of the paper is organized as follows. In Section 2 we introduce HTL and provide necessary backgrounds for KS and KRR. We formalize our transformation function based framework in Section 3. Our main theoretical results are in Section 4 and specifically in Section 4.1 and Section 4.2 we provide explicit risk bounds for KS and KRR, respectively. In Section 5 we analyze cross-validation in HTL setting and in Section 6 we conduct experiments on real world data data. We conclude with a brief discussion of avenues for future work.

## 2 Preliminaries

### 2.1 Problem Setup

In this paper, we assume both $X \in \mathbb{R}^d$ and $Y \in \mathbb{R}$ lie in compact subsets: $||X||_2 \leq \triangle_X$, $|Y| \leq \triangle_Y$ for some $\triangle_X, \triangle_Y \in \mathbb{R}_+$. Throughout the paper, we use $\mathcal{T} = \{(X_i, Y_i)\}_{i=1}^n$ to denote a set of samples. Let $(X^{so}, Y^{so})$ be the sample from the source domain, and $(X^{ta}, Y^{ta})$ the sample from the target domain. In our setting, there are $n_{so}$ samples drawn i.i.d from the source distribution: $\mathcal{T}^{so} = \{(X_i^{so}, Y_i^{so})\}_{i=1}^{n_{so}}$, and $n_{ta}$ samples drawn i.i.d from the target distribution: $\mathcal{T}^{ta} = \{(X_i^{ta}, Y_i^{ta})\}_{i=1}^{n_{ta}}$. In addition, we also use $n_{val}$ samples drawn i.i.d from the target domain for cross-validation. We model the joint relation between $X$ and $Y$ by: $Y^{so} = f^{so}(X^{so}) + \epsilon^{so}$ and $Y^{ta} = f^{ta}(X^{ta}) + \epsilon^{ta}$ where $f^{so}$ and $f^{ta}$ are regression functions and we assume the noise $\mathbf{E}[\epsilon^{so}] = \mathbf{E}[\epsilon^{ta}] = 0$, i.i.d,

and bounded. We use $\mathcal{A} : \mathcal{T} \to \hat{f}$ to denote an algorithm that takes a set of samples and produce an estimator. Given an estimator $\hat{f}$, we define the integrated $L_2$ risk as $R(\hat{f}) = \mathbf{E}\left[\left(\hat{f}(X) - Y\right)^2\right]$ where the expectation is taken over the distribution of $(X, Y)$. Similarly, the empirical $L_2$ risk on a set of sample $\mathcal{T}$ is defined as $\hat{R}(\hat{f}) = \frac{1}{n}\sum_{i=1}^{n}\left(Y_i - \hat{f}(X_i)\right)^2$. In HTL setting, we use $\hat{f}^{so}$ an estimator from the source domain to facilitate the learning procedure for $f^{ta}$.

## 2.2 Kernel Smoothing

We say a function $f$ is in the $(\lambda, \alpha)$ Hölder class [Wasserman, 2006], if for any $x, x' \in \mathbb{R}^d$, $f$ satisfies $|f(x) - f(x')| \leq \lambda \|x - x'\|_2^\alpha$, for some $\alpha \in (0, 1)$. The kernel smoothing method uses a positive kernel $K$ on $[0, 1]$, highest at 0, decreasing on $[0, 1]$, 0 outside $[0, 1]$, and $\int_{\mathbb{R}^d} u^2 K(u) < \infty$. Using $\mathcal{T} = \{(X_i, Y_i)\}_{i=1}^n$, the kernel smoothing estimator is defined as follows: $\hat{f}(x) = \sum_{i=1}^{n} w_i(x)Y_i$, where $w_i(x) = \frac{K(\|x - X_i\|/h)}{\sum_{j=1}^{n} K(\|x - X_j\|/h)} \in [0, 1]$.

## 2.3 Kernel Ridge Regression

Another popular non-linear estimator is the kernel ridge regression (KRR) which uses the theory of reproducing kernel Hilbert space (RKHS) for regression [Vovk, 2013]. Any symmetric positive semidefinite kernel function $K : \mathcal{R}^d \times \mathcal{R}^d \to \mathcal{R}$ defines a RKHS $\mathcal{H}$. For each $x \in \mathcal{R}^d$, the function $z \to K(z, x)$ is contained in the Hilbert space $\mathcal{H}$; moreover, the Hilbert space is endowed with an inner product $\langle \cdot, \cdot \rangle_{\mathcal{H}}$ such that $K(\cdot, x)$ acts as the kernel of the evaluation functional, meaning $\langle f, K(x, \cdot) \rangle_{\mathcal{H}} = f(x)$ for $f \in \mathcal{H}$. In this paper we assume $K$ is bounded: $\sup_{x \in \mathbb{R}^d} K(x, x) = k < \infty$. Given the inner product, the $\mathcal{H}$ norm of a function $g \in \mathcal{H}$ is defined as $\|g\|_{\mathcal{H}} \triangleq \sqrt{\langle g, g \rangle}_{\mathcal{H}}$ and similarly the $L_2$ norm, $\|g\|_2 \triangleq \left(\int_{\mathbb{R}^d} g(x)^2 dP_X\right)^{1/2}$ for a given $P_X$. Also, the kernel induces an integral operator $T_K : L_2(P_X) \to L_2(P_X)$: $T_K[f](x) = \int_{\mathbb{R}^d} K(x', x) f(x') dP_x(x')$ with countably many non-zero eigenvalues: $\{\mu_i\}_{i \geq 1}$. For a given function $f$, the approximation error is defined as: $A_f(\lambda) \triangleq \inf_{h \in \mathcal{H}}\left(\|h - f\|_{L_2(P_X)}^2 + \lambda \|h\|_{\mathcal{H}}^2\right)$ for $\lambda \geq 0$. Finally the estimated function evaluated at point $x$ can be written as $\hat{f}(x) = K(\mathbf{X}, x)(K(\mathbf{X}, \mathbf{X}) + n\lambda I)^{-1}\mathbf{Y}$ where $\mathbf{X} \in \mathbb{R}^{n \times d}$ are the inputs of training samples and $\mathbf{Y} \in \mathbb{R}^{n \times 1}$ are the training labels Vovk [2013].

## 2.4 Related work

Before we present our framework, it is helpful to give a brief overview of existing literature on theoretical analysis of transfer learning. Many previous works focused on the settings when only *unlabeled* data from the target domain are available [Huang et al., 2006, Sugiyama et al., 2008, Yu and Szepesvári, 2012]. In particular, a line of research has been established based on distribution discrepancy, a loss induced metric for the source and target distributions [Mansour et al., 2009, Ben-David et al., 2007, Blitzer et al., 2008, Cortes and Mohri, 2011, Mohri and Medina, 2012]. For example, recently Cortes and Mohri [2014] gave generalization bounds for kernel based methods under convex loss in terms of discrepancy.

In many real world applications such as yield prediction from pictures [Nuske et al., 2014], or prediction of response time from fMRI [Verstynen, 2014], some labeled data from the target domain is also available. Cortes et al. [2015] used these data to improve their discrepancy minimization algorithm. Zhang et al. [2013] focused on modeling target shift ($P(Y)$ changes), conditional shift ($P(X|Y)$ changes), and a combination of both. Recently, Wang and Schneider [2014] proposed a kernel mean embedding method to match the conditional probability in the kernel space and later derived generalization bound for this problem Wang and Schneider [2015]. Kuzborskij and Orabona [2013, 2016], Kuzborskij et al. [2016] gave excess risk bounds for target domain estimator in the form of a linear combination of estimators from multiple source domains and an additional linear function. Ben-David and Urner [2013] showed a similar bound of the same setting with different quantities capturing the relatedness. Wang et al. [2016] showed that if the features of source and target domain are $[0, 1]^d$, using orthonormal basis function estimator, transfer learning achieves better excess risk

guarantee if $f^{ta} - f^{so}$ can be approximated by the basis functions easier than $f^{ta}$. Their work can be viewed as a special case of our framework using the transformation function $G(a, b) = a + b$.

# 3 Transformation Functions

In this section, we first define our class of models and give a meta-algorithm to learn the target regression function. Our models are based on the idea that transfer learning is helpful when one transforms the target domain regression problem into a simpler regression problem using source domain knowledge. Consider the following example.

**Example: Offset Transfer.** Let $f^{so}(x) = \sqrt{x(1-x)} \sin\left(\frac{2.1\pi}{x+0.05}\right)$ and $f^{ta}(x) = f^{so}(x) + x$. $f^{so}$ is the so called Doppler function. It requires a large number of samples to estimate well because of its lack of smoothness Wasserman [2006]. For the same reason, $f^{ta}$ is also difficult to estimate directly. However, if we have enough data from the source domain, we can have a fairly good estimate of $f^{so}$. Further, notice that the offset function $w(x) = f^{ta}(x) - f^{so}(x) = x$, is just a linear function. Thus, instead of directly using $\mathcal{T}^{ta}$ to estimate $f^{ta}$, we can use the target domain samples to find an estimate of $w(x)$, denoted by $\hat{w}(x)$, and our estimator for the target domain is just: $\hat{f}^{ta}(x) = \hat{f}^{so}(x) + \hat{w}(x)$. Figure 1b shows this technique gives improved fitting for $f^{ta}$.

The previous example exploits the fact that function $w(x) = f^{ta}(x) - f^{so}(x)$ is a simpler function than $f^{ta}$. Now we generalize this idea further. Formally, we define the *transformation function* as $G(a, b) : \mathbb{R}^2 \to \mathbb{R}$ where we assume that given $a \in \mathbb{R}$, $G(a, \cdot)$ is invertible. Here $a$ will be the regression function of the source domain evaluated at some point and the output of $G$ will be the regression function of the target domain evaluated at the same point. Let $G_a^{-1}(\cdot)$ denote the inverse of $G(a, \cdot)$ such that $G\left(a, G_a^{-1}(c)\right) = c$. For example if $G(a, b) = a + b$ and $G_a^{-1}(c) = c - a$. For a given $G$ and a pair $(f^{so}, f^{ta})$, they together induce a function $w_G(x) = G_{f^{so}(x)}^{-1}(f^{ta}(x))$. In the offset transfer example, $w_G(x) = x$. By this definition, for any $x$, we have $G\left(f^{so}(x), w_G(x)\right) = f^{ta}(x)$. We call $w_G$ the *auxiliary function* of the transformation function $G$. In the HTL setting, $G$ is a user-defined transformation that represents users' prior knowledge on the relation between the source and target domains. Now we list some other examples:

**Example: Scale-Transfer.** Consider $G(a, b) = ab$. This transformation function is useful when $f^{so}$ and $f^{ta}$ satisfy a smooth scale transfer. For example, if $f^{ta} = cf^{so}$, for some constant $c$, then $w_G(x) = c$ because $f^{ta}(x) = G\left(f^{so}(x), w_G(x)\right) = f^{so}(x) w_G(x) = f^{so}(x) c$. See Figure 1c.

**Example: Non-Transfer.** Consider $G(a, b) = b$. Notice that $f^{ta}(x) = w_G(x)$ and so $f^{so}$ is irrelevant. Thus this model is equivalent to traditional regression on the target domain since data from the source domain does not help.

## 3.1 A Meta Algorithm

Given the transformation $G$ and data, we provide a general procedure to estimate $f^{ta}$. The spirit of the algorithm is turning learning a complex function $f^{ta}$ into an easier function $w_G$. First we use an algorithm $\mathcal{A}_{so}$ that takes $\mathcal{T}^{so}$ to obtain $\hat{f}^{so}$. Since we have sufficient data from the source domain, $\hat{f}^{so}$ should be close to the true regression function $f^{so}$. Second, we construct a new data set using the $n_{ta}$ data points from the target domain: $\mathcal{T}^{w_G} = \left\{\left(X_i^{ta}, H_G\left(\hat{f}^{so}(X_i^{ta}), Y_i^{ta}\right)\right)\right\}_{i=1}^{n_{ta}}$ where $H_G : \mathbb{R}^2 \to \mathbb{R}$ and satisfies

$$\mathbf{E}\left[H_G\left(f^{so}(X_i^{ta}), Y_i^{ta}\right)\right] = G_{f^{so}(X_i^{ta})}^{-1}\left(f^{ta}(X_i^{ta})\right) = w_G(X_i^{ta})$$

where and the expectation is taken over $\epsilon_{ta}$. Thus, we can use these newly constructed data to learn $w_G$ with algorithm $\mathcal{A}_{W_G}$: $\hat{w}_G = \mathcal{A}_{W_G}\left(\mathcal{T}^{W_G}\right)$. Finally, we plug trained $\hat{f}^{so}$ and $\hat{w}_G$ into transformation $G$ to obtain an estimation for $f^{ta}$: $\hat{f}^{ta}(X) = G(\hat{f}^{so}(X), \hat{w}_G(X))$. Pseudocode is shown in Algorithm 1.

**Unbiased Estimator** $H_G\left(f^{so}(X^{ta}), Y^{ta}\right)$: In Algorithm 1, we require an unbiased estimator for $w_G(X^{ta})$. Note that if $G(a, b)$ is linear $b$ or $\epsilon^{ta} = 0$, we can simply set $H_G(f^{so}(X), Y) = G_{f^{so}(X)}^{-1}(Y)$. For other scenarios, $G_{f^{so}(X_i^{ta})}^{-1}(Y_i^{ta})$ is biased: $\mathbf{E}\left[G_{f^{so}(X_i^{ta})}^{-1}(Y_i^{ta})\right] \neq G_{f^{so}(x)}^{-1}(f^{ta}(x))$ and we need to design estimator using the structure of $G$.

---
**Algorithm 1** Transformation Function based Transfer Learning
---
**Inputs:** Source domain data: $\mathcal{T}^{so} = \{(X_i^{so}, Y_i^{so})\}_{i=1}^{n_{so}}$, target domain data: $\mathcal{T}^{ta} = \{(X_i^{ta}, Y_i^{ta})\}_{i=1}^{n_{ta}}$, transformation function: $G$, algorithm to train $f^{so}$: $\mathcal{A}_{so}$, algorithm to train $w_G$: $\mathcal{A}_{w_G}$ and $H_G$ an unbiased estimator for estimating $w_G$.

**Outputs:** Regression function for the target domain: $\hat{f}^{ta}$.

1: Train the source domain regression function $\hat{f}^{so} = \mathcal{A}_{so}(\mathcal{T}^{so})$.
2: Construct new data using $\hat{f}^{so}$ and $\mathcal{T}^{ta}$: $\mathcal{T}^{w_G} = \{(X_i^{ta}, W_i)\}_{i=1}^{n_{ta}}$, where $W_i = H_G\left(\hat{f}^{so}(X_i^{ta}), Y_i^{ta}\right)$.
3: Train the auxiliary function: $\hat{w}_G = \mathcal{A}_{W_G}(\mathcal{T}^{w_G})$.
4: Output the estimated regression for the target domain: $\hat{f}^{ta}(X) = G\left(\hat{f}^{so}(X), \hat{w}_G(X)\right)$.
---

**Remark 1:** Many transformation functions are equivalent to a transformation function $G'(a, b)$ where $G'(a, b)$ is linear in $b$. For example, for $G(a, b) = ab^2$, i.e., $f^{ta}(x) = f^{so}(x) w_G^2(x)$, consider $G'(a, b) = ab$ where $b$ in $G'$ stands for $b^2$ in $G$, i.e., $f^{ta}(x) = f^{so}(x) w'_G(x)$. Therefore $w'_G = w_G^2$ and we only need to estimate $w'_G$ well instead of estimating $w_G$. More generally, if $G(a, b)$ can be factorized as $G(a, b) = g_1(a) g_2(b)$, i.e., $f^{ta}(x) = g_1(f^{so}(x)) g_2(w_G(x))$, we only need to estimate $g_2(w_G(x))$ and the convergence rate depends on the structure of $g_2(w_G(x))$.

**Remark 2:** When $G$ is not linear in $b$ and $\epsilon^{ta} \neq 0$, observe that in Algorithm 1, we treat $Y_i^{ta}$s as noisy covariates to estimate $w_G(X_i)$s. This problem is called error-in-variable or measurement error and has been widely studied in statistics literature. For details, we refer the reader to the seminal work by Carroll et al. [2006]. There is no universal estimator for the measurement error problem. In Section B, we provide a common technique, regression calibration to deal with measurement error problem.

## 4 Excess Risk Analyses

In this section, we present theoretical analyses for the proposed class of models and estimators. First, we need to impose some conditions on $G$. The first assures that if the estimations of $f^{so}$ and $w_G$ are close to the source regression and auxiliary function, then our estimator for $f^{ta}$ is close to the true target regression function. The second assures that we are estimating a regular function.

**Assumption 1** $G(a, b)$ is $L$-Lipschitz: $|G(a, b) - G(a', b')| \leq L \|(a, b) - (a', b')\|_2$ and is invertible with respect to $b$ given $a$, i.e. if $G(x, y) = z$ then $G_x^{-1}(z) = y$.

**Assumption 2** Given $G$, the induced auxiliary function $w_G$ is bounded: for $x : \|x\|_2 \leq \triangle_X$, $w_G(x) \leq B$ for some $B > 0$.

Offset Transfer and Non-Transfer satisfy these conditions with $L = 1$ and $B = \triangle_Y$. Scale Transfer satisfies these assumptions when $f^{so}$ is lower bounded from away 0. Lastly, we assume our unbiased estimator is also regular.

**Assumption 3** For $x : \|x\|_2 \leq \triangle_X$ and $y : |y| \leq \triangle_Y$, $H_G(x, y) \leq B$ for some $B > 0$ and $H_G$ is Lipschitz continuous in the first argument: $|H_G(x, y) - H_G(x', y)| \leq L|x - x'|$ for some $L > 0$.

We begin with a general result which only requires the stability of $\mathcal{A}_{W_G}$:

**Theorem 1** Suppose for any two sets of samples that have same features but different labels: $\mathcal{T} = \{(X_i^{ta}, W_i)\}_{i=1}^{n_{ta}}$ and $\widetilde{\mathcal{T}} = \left\{\left(X_i^{ta}, \widetilde{W}_i\right)\right\}_{i=1}^{n_{ta}}$, the algorithm $\mathcal{A}_{w_G}$ for training $w_G$ satisfies:

$$\left\|\mathcal{A}_{w_G}(\mathcal{T}) - \mathcal{A}_{w_G}\left(\widetilde{\mathcal{T}}\right)\right\|_\infty \leq \sum_{i=1}^{n_{ta}} c_i(X_i^{ta}) \left|W_i - \widetilde{W}_i\right|, \tag{1}$$

where $c_i$ only depends on $X_i^{ta}$. Then for any $x$,

$$\left|\hat{f}^{ta}(x) - f^{ta}(x)\right|^2 = O\left(\left|\hat{f}^{so}(x) - f^{so}(x)\right|^2 + |\widetilde{w}_G(x) - w_G(x)|^2 + \right.$$

$$\left.\left(\sum_{i=1}^{n_{ta}} c_i\left(X_i^{ta}\right)\left|\hat{f}^{so}\left(X_i^{ta}\right) - f^{so}\left(X_i^{ta}\right)\right|\right)^2\right)$$

where $\tilde{w}_G = \mathcal{A}_{w_G}\left(\{(X_i^{ta}, H_G\left(f^{so}\left(X_i^{ta}\right), Y_i^{ta}\right))\}_{i=1}^{n_{ta}}\right)$, the estimated auxiliary function trained based on true source domain regression function.

Theorem 1 shows how the estimation error in the source domain function propagates to our estimation of the target domain function. Notice that if we happen to know $f^{so}$, then the error is bounded by $O\left(|\tilde{w}_G(x) - w_G(x)|^2\right)$, the estimation error of $w_G$. However, since we are using estimated $f^{so}$ to construct training samples for $w_G$, the error might accumulate as $n_{ta}$ increases. Though the third term in Theorem 1 might increase with $n_{ta}$, it also depends on the estimation error of $f^{so}$ which is relatively small because of the large amount of source domain data.

The stability condition (1) we used is related to the uniform stability introduced by Bousquet and Elisseeff Bousquet and Elisseeff [2002] where they consider how much will the output change if one of the training instance is removed or replaced by another whereas ours depends on two different training data sets. The connection between transfer learning and stability has been discovered by Kuzborskij and Orabona [2013], Liu et al. [2016] and Zhang [2015] in different settings, but they only showed bounds for generalization, not for excess risk.

## 4.1 Kernel Smoothing

We first analyze kernel smoothing method.

**Theorem 2** *Suppose the support of $X^{ta}$ is a subset of the support of $X^{so}$ and the probability density of $P_{X^{so}}$ and $P_{X^{ta}}$ are uniformly bounded away from below on their supports. Further assume $f^{so}$ is $(\lambda_{so}, \alpha_{so})$ Hölder and $w_G$ is $(\lambda_{w_G}, \alpha_{w_G})$ Hölder . If we use kernel smoothing estimation for $f^{so}$ and $w_G$ with bandwidth $h_{so} \asymp n_{so}^{-1/(2\alpha_{so}+d)}$ and $h_{w_G} \asymp n_{ta}^{-1/(2\alpha_{w_G}+d)}$, with probability at least $1 - \delta$ the risk satisfies:*

$$\mathbf{E}\left[R\left(\hat{f}^{ta}\right)\right] - R\left(f^{ta}\right) = O\left(n_{so}^{\frac{-2\alpha_{so}}{2\alpha_{so}+d}} + n_{ta}^{\frac{-2\alpha_{w_G}}{2\alpha_{w_G}+d}}\right)\log\left(\frac{1}{\delta}\right)$$

*where the expectation is taken over $\mathcal{T}^{so}$ and $\mathcal{T}^{ta}$.*

Theorem 2 suggests that the risk depends on two sources, one from estimation of $f^{so}$ and one from estimation of $w_G$. For the first term, since in the typical transfer learning scenarios $n_{so} >> n_{ta}$, it is relatively small in the setting we focus on. The second terms shows the power of transfer learning on transforming a possibly complex target regression function into a simpler auxiliary function. It is well known that learning $f^{ta}$ only using target domain has risk of the order $\Omega\left(n_{ta}^{-2\alpha_{f^{ta}}/\left(2\alpha_{f^{ta}}+d\right)}\right)$. Thus, if the auxiliary function is smoother than the target regression function, i.e. $\alpha_{w_G} > \alpha_{f^{ta}}$, we obtain better statistical rate.

## 4.2 Kernel Ridge Regression

Next, we give an upper bound for the excess risk using KRR:

**Theorem 3** *Suppose $P_{X^{so}} = P_{X^{ta}}$ and the eigenvalues of the integral operator $T_K$ satisfy $\mu_i \leq ai^{-1/p}$ for $i \geq 1$ $a \geq 16\triangle_Y^4$, $p \in (0,1)$ and there exists a constant $C \geq 1$ such that for $f \in \mathcal{H}$, $||f||_\infty \leq C ||f||_{\mathcal{H}}^p \cdot ||f||_{L_2(P_X)}^{1-p}$. Furthur assume that $A^{f^{so}}(\lambda) \leq c\lambda^{\beta_{so}}$ and $A^{w_G}(\lambda) \leq c\lambda^{\beta_{w_G}}$. If we use KRR for estimating $f^{so}$ and $w_G$ with regularization parameters $\lambda_{so} \asymp n_{so}^{-1/(\beta_{so}+p)}$ and*

$\lambda_{w_G} \asymp n_{ta}^{-1/(\beta_{w_G}+p)}$, *then with probability at least $1 - \delta$ the excess risk satisfies:*

$$\mathbf{E}\left[R\left(\hat{f}^{ta}\right)\right] - R\left(f^{ta}\right) = O\left(\left(n_{ta}^{\frac{2}{\beta_{w_G}+p}}\log\left(n_{ta}\right)\cdot n_{so}^{\frac{-\beta_{so}}{\beta_{so}+p}} + n_{ta}^{\frac{-\beta_{w_G}}{\beta_{w_G}+p}}\right)\log\left(\frac{1}{\delta}\right)\right)$$

*where the expectation is taken over $\mathcal{T}^{so}$ and $\mathcal{T}^{ta}$.*

Similar to Theorem 2, Theorem 3 suggests that the estimation error comes from two sources. For estimating the auxiliary function $w_G$, the statistical rate depends on properties of the kernel induced RKHS, and how far the auxiliary function is from this space. For the ease of presentation, we assume $P_{X^{so}} = P_{X^{ta}}$, so the approximation errors $A^{f^{so}}$ and $A^{f^{ta}}$ are defined on the same domain. The error of estimating $f^{so}$ is amplified by $O\left(\lambda_{w_G}^{-2}\log\left(n_{ta}\right)\right)$, which is worse than that of nonparametric kernel smoothing. We believe this $\lambda_{w_G}^{-2}$ is nearly tight because Bousquet and Elisseeff have shown the uniform algorithmic stability parameter for KRR is $O\left(\lambda_{w_G}^{-2}\right)$ Bousquet and Elisseeff [2002]. Steinwart et al. Steinwart et al. [2009] showed that for non-transfer learning, the optimal statistical rate for excess risk is $\Omega\left(n_{ta}^{\frac{-\beta_{ta}}{\beta_{ta}+p}}\right)$, so if $\beta_{wg} \geq \beta_{ta}$ and $n_{so}$ is sufficiently large then we achieve improved convergence rate through transfer learning.

**Remark:** Theorem 2 and 3 are not directly comparable because our assumptions on the function spaces of these two theorems are different. In general, Hölder space is only a Banach space but not a Hilbert space. We refer readers to Theorem 1 in Zhou [2008] for details.

## 5 Finding the Best Transformation Function

In the previous section we showed for a specific transformation function $G$, if auxiliary function is smoother than the target regression function then we have smaller excess risk. In practice, we would like to try out a class of transformation functions $\mathcal{G}$, which is possibly uncountable. We can construct a subset of $\overline{\mathcal{G}} \subset \mathcal{G}$, which is finite and satisfies that each $G$ in $\mathcal{G}$ there is a $\overline{G}$ in $\overline{\mathcal{G}}$ that is close to $G$. Here we give an example. Consider the transformation functions that have the form: $\mathcal{G} = \{G(a,b) = \alpha a + b \text{ where } |\alpha| \leq L_\alpha, |a| \leq L_a\}$. We can quantize this set of transformation functions by considering a subset of $\mathcal{G}$: $\overline{\mathcal{G}} = \{G(a,b) = k\epsilon a + b\}$ where $\epsilon = \frac{L_\alpha}{2K}, k = -K, \cdots, 0, \cdots, K$ and $|a| \leq L_a$. Here $\epsilon$ is the quantization unit.

The next theorem shows we only need to search the transformation function $\overline{G}$ in $\overline{\mathcal{G}}$ whose corresponding estimator $\hat{f}^{ta}_{\overline{G}}$ has the lowest empirical risk on the validation dataset.

**Theorem 4** *Let $\mathcal{G}$ be a class of transformation functions and $\overline{\mathcal{G}}$ be its $\|\cdot\|_\infty$ norm $\epsilon$-cover. Suppose $w_G$ satisfies the same assumption in Theorem 1 and for any two $G_1, G_2 \in \mathcal{G}$, $\|w_{G_1} - w_{G_2}\|_\infty \leq L \|G_1 - G_2\|_\infty$ for some constant $L$. Denote $G^\star = \operatorname{argmin}_{G \in \mathcal{G}} R\left(\hat{f}^{ta}_G\right)$ and $\overline{G}^\star = \operatorname{argmin}_{G \in \overline{\mathcal{G}}} \hat{R}\left(\hat{f}^{ta}_G\right)$. If we choose $\epsilon = O\left(\frac{R\left(\hat{f}^{ta}_{G^\star}\right)}{\sum_{i=1}^{n_{ta}} ci}\right)$ and $n_{val} = \Omega\left(\log\left(\left|\overline{\mathcal{G}}\right|/\delta\right)\right)$, the with probability at least $1 - \delta$, $\mathbf{E}\left[R\left(\hat{f}^{ta}_{\overline{G}^\star}\right)\right] - R\left(f^{ta}\right) = O\left(\mathbf{E}\left[R\left(\hat{f}^{ta}_{G^\star}\right)\right] - R\left(f^{ta}\right)\right)$ where the expectation is taken over $\mathcal{T}^{so}$ and $\mathcal{T}^{ta}$.*

**Remark 1:** This theorem implies that if no-transfer function ($G(a,b) = b$) is in $\mathcal{G}$ then we will end up choosing a transformation function that has the same order of excess risk as using no-transfer learning algorithm, thus avoiding negative transfer.

**Remark 2:** Note number of validation set is only logarithmically depending on the size of set of transformation functions. Therefore, we only need to use a very small amount of data from the target domain to do cross-validation.

## 6 Experiments

In this section we use robotics and neural imaging data to demonstrate the effectiveness of the proposed framework. We conduct experiments on real-world data sets with the following procedures.

|                 | $n_{ta}=10$        | $n_{ta}=20$        | $n_{ta}=40$        | $n_{ta}=80$        | $n_{ta}=160$       | $n_{ta}=320$       |
|-----------------|--------------------|--------------------|--------------------|--------------------|--------------------|--------------------|
| Only Target KS  | $0.086 \pm 0.022$  | $0.076 \pm 0.010$  | $0.066 \pm 0.008$  | $0.064 \pm 0.007$  | $0.065 \pm 0.006$  | $0.063 \pm 0.005$  |
| Only Target KRR | $0.080 \pm 0.017$  | $0.078 \pm 0.022$  | $0.063 \pm 0.013$  | $0.050 \pm 0.007$  | $0.048 \pm 0.006$  | $\mathbf{0.040 \pm 0.005}$ |
| Only Source KRR | $0.098 \pm 0.017$  | $0.098 \pm 0.017$  | $0.098 \pm 0.017$  | $0.098 \pm 0.017$  | $0.098 \pm 0.017$  | $0.098 \pm 0.017$  |
| Combined KS     | $0.092 \pm 0.011$  | $0.084 \pm 0.008$  | $0.077 \pm 0.009$  | $0.075 \pm 0.006$  | $0.074 \pm 0.006$  | $0.067 \pm 0.006$  |
| Combined KRR    | $0.087 \pm 0.025$  | $0.077 \pm 0.015$  | $0.062 \pm 0.009$  | $0.061 \pm 0.005$  | $0.047 \pm 0.003$  | $0.041 \pm 0.004$  |
| CDM             | $0.105 \pm 0.023$  | $0.074 \pm 0.020$  | $0.064 \pm 0.008$  | $0.060 \pm 0.007$  | $0.053 \pm 0.009$  | $0.056 \pm 0.004$  |
| Offset KS       | $0.080 \pm 0.026$  | $0.066 \pm 0.023$  | $\mathbf{0.052 \pm 0.006}$ | $0.054 \pm 0.006$ | $0.050 \pm 0.003$ | $0.052 \pm 0.004$ |
| Offset KRR      | $0.146 \pm 0.112$  | $0.066 \pm 0.017$  | $0.053 \pm 0.007$  | $\mathbf{0.048 \pm 0.006}$ | $\mathbf{0.043 \pm 0.004}$ | $0.041 \pm 0.003$ |
| Scale KS        | $\mathbf{0.078 \pm 0.022}$ | $\mathbf{0.065 \pm 0.013}$ | $0.056 \pm 0.009$ | $0.056 \pm 0.005$ | $0.054 \pm 0.008$ | $0.055 \pm 0.004$ |
| Scale KRR       | $0.102 \pm 0.033$  | $0.095 \pm 0.100$  | $0.057 \pm 0.014$  | $0.052 \pm 0.010$  | $0.044 \pm 0.004$  | $0.042 \pm 0.002$  |

Table 1: 1 standard deviation intervals for the mean squared errors of various algorithms when transferring from kin-8fm to kin-8nh. The values in bold are the smallest errors for each $n_{ta}$. Only Source KS has much worse performance than other algorithms so we do not show its result here.

- Directly training on the target data $\mathcal{T}^{ta}$ (Only Target KS, Only Target KRR).
- Only training on the source data $\mathcal{T}^{so}$ (Only Source KS, Only Source KRR).
- Training on the combined source and target data (Combined KS, Combined KRR).
- The CDM algorithm proposed by Wang and Schneider [2014] with KRR (CDM).
- The algorithm described in this paper with $G(a,b) = (a+\alpha)b$ where $\alpha$ is a hyper-parameter (Scale KS, Scale KRR).
- The algorithm described in this paper with $G(a,b) = \alpha a + b$ where $\alpha$ is a hyper-parameter (Offset KS, Offset KRR). $\angle$

For the first experiment, we vary the size of the target domain to study the effect of $n_{ta}$ relative to $n_{so}$. We use two datasets from the 'kin' family in Delve [Rasmussen et al., 1996]. The two datasets we use are 'kin-8fm' and 'kin-8nh', both with 8 dimensional inputs. kin-8fm has fairly linear output, and low noise. kin-8nh on the other hand has non-linear output, and high noise. We consider the task of transfer learning from kin-8fm to kin-8nh. In this experiment, We set $n_{so}$ to 320, and vary $n_{ta}$ in $\{10, 20, 40, 80, 160, 320\}$. Hyper-parameters were picked using grid search with 10-fold cross-validation on the target data (or source domain data when not using the target domain data).

Table 1 shows the mean squared errors on the target data. To better understand the results, we show a box plot of the mean squared errors for $n_{ta} = 40$ onwards in Figure 2(a). The results for $n_{ta} = 10$ and $n_{ta} = 20$ have high variance, so we do not show them in the plot. We also omit the results of Only Source KRR because of its poor performance. We note that our proposed algorithm outperforms other methods across nearly all values of $n_{ta}$ especially when $n_{ta}$ is small. Only when there are as many points in the target as in the source, does simply training on the target give the best performance. This is to be expected since the primary purpose in doing transfer learning is to alleviate the problem of lack of data in the target domain. Though quite comparable, the performance of the scale methods was worse than the offset methods in this experiment. In general, we would use cross-validation to choose between the two.

We now consider another real-world dataset where the covariates are fMRI images taken while subjects perform a Stroop task [Stroop, 1935]. We use the dataset collected by Verstynen [2014] which contains fMRI data of 28 subjects. A total of 120 trials were presented to each participant and fMRI data was collected throughout the trials, and went through a standard post-processing scheme. The result of this is a feature vector corresponding to each trial that describes the activity of brain regions (voxels), and the goal is to use this to predict the response time.

To frame the problem in the transfer learning setting, we consider as source the data of all but one subject. The goal is to predict on the remaining subject. We performed five repetitions for each algorithm by drawing $n_{so} = 300$ data points randomly from the 3000 points in the source domain. We used $n_{ta} = 80$ points from the target domain for training and cross-validation; evaluation was done on the 35 remaining points in the target domain. Figure 2 (b) shows a box plot of the coeffecient of determination values (R-squared) for the best performing algorithms. R-squared is defined as $1 - SS_{res}/SS_{tot}$ where $SS_{res}$ is the sum of squared residuals, and $SS_{tot}$ is the total sum of squares. Note that R-squared can be negative when predicting on unseen samples – which were not used to fit the model – as in our case. When positive, it indicates the proportion of explained variance in the dependent variable (higher the better). From the plot, it is clear that Offset KRR and Only Target KRR have the best performances on average and Offset KRR has smaller variance.

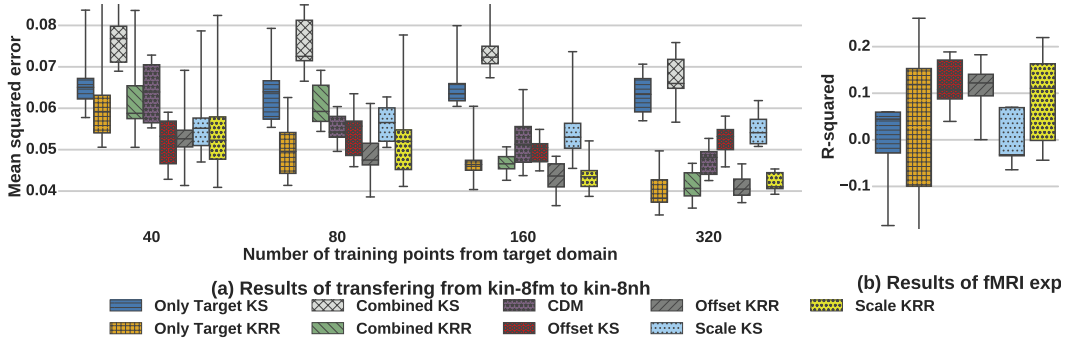

Figure 2: Box plots of experimental results on real datasets. Each box extends from the first to third quartile, and the horizontal lines in the middle are medians. For the robotics data, we report mean squared error (lower the better) and for the fMRI data, we report R-squared (the higher the better). For the ease of presentation, we only show results of algorithms with good performances.

|  | Mean | Median | Standard Deviation |
|---|---|---|---|
| Only Target KS | -0.0096 | 0.0444 | 0.1041 |
| Only Target KRR | 0.1041 | 0.1186 | 0.2361 |
| Only Source KS | -0.4932 | -0.5366 | 0.4555 |
| Only Source KRR | -0.8763 | -0.9363 | 0.6265 |
| Combined KS | -0.7540 | -0.2023 | 1.5109 |
| Combined KRR | -0.5868 | -0.0691 | 1.3223 |
| CDM | -3.1183 | -3.4510 | 2.6473 |
| Offset KS | **0.1190** | 0.1081 | **0.0612** |
| Offset KRR | 0.1080 | **0.1221** | 0.0682 |
| Scale KS | 0.0017 | -0.0321 | 0.0632 |
| Scale KRR | 0.0897 | 0.1107 | 0.1104 |

Table 2: Mean, median, and standard deviation for the coefficient of determination (R-squared) of various algorithms on the fMRI dataset.

Table 2 shows the full table of results for the fMRI task. Using only the source data produces large negative R-squared, and while Only Target KRR does produce a positive mean R-squared, it comes with a high variance. On the other hand, both Offset methods have low variance, showing consistent performance. For this particular case, the Scale methods do not perform as well as the Offset methods, and as has been noted earlier, in general we would use cross validation to select an appropriate transfer function.

## 7 Conclusion and Future Works

In this paper, we proposed a general transfer learning framework for the HTL regression problem when there is some data available from the target domain. Theoretical analysis shows it is possible to achieve better statistical rate using transfer learning than standard supervised learning.

Now we list two future directions and how our results could be further improved. First, in many real world applications, there is also a large amount of *unlabeled* data from the target domain available. Combining our proposed framework with previous works for this scenario [Cortes and Mohri, 2014, Huang et al., 2006] is a promising direction to pursue. Second, we only present upper bounds in this paper. It is an interesting direction to obtain lower bounds for HTL and other transfer learning scenarios.

## 8 Acknowledgements

S.S.D. and B.P. were supported by NSF grant IIS1563887 and ARPA-E Terra program. A.S. was supported by AFRL grant FA8750-17-2-0212.

## Footnotes

[1]We formally define the transformation functions in Section 3.

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
