[Supplementary Material · transfer_supp.pdf]

# A  Proofs

## A.1  Proof of Theorem 1

*Proof of Theorem 1.* The proof just uses assumptions on the transformation function and stability of the training algorithm.

$$\left| \hat{f}^{ta}(x) - f^{ta}(x) \right|^2$$

$$= \left| G\left(\hat{f}^{so}(x), \hat{w}_G(x)\right) - G\left(f^{so}(x), w_G(x)\right) \right|^2 \tag{2}$$

$$\leq L^2 \left| \hat{f}^{so}(x) - f^{so}(x) \right|^2 + L^2 \left| \hat{w}_G(x) - w_G(x) \right|^2 \tag{3}$$

$$\leq L^2 \left| \hat{f}^{so}(x) - f^{so}(x) \right|^2 + 2L^2 \left| \hat{w}_G(x) - \tilde{w}_G(x) \right|^2 + 2L^2 \left| \tilde{w}_G(x) - w_G(x) \right|^2 \tag{4}$$

$$\leq L^2 \left| \hat{f}^{so}(x) - f^{so}(x) \right|^2 + 2L^2 \left( \sum_{i=1}^{n_{ta}} c_i\left(X_i^{ta}\right) \left| W_i - \widetilde{W}_i \right| \right)^2 + 2L^2 \left| \tilde{w}_G(x) - w_G(x) \right|^2 \tag{5}$$

where (2) is by the requirement of $G$, (3) is by the Lipschitz condition of $G$, (4) is because $(a-b)^2 \leq 2(a-c)^2 + 2(c-b)^2$ and (5) is by our stability assumption of $\mathcal{A}_{w_G}$. Now, we are left bounding $\left( \sum_{i=1}^{n_{ta}} c_i \left| W_i - \widetilde{W}_i \right| \right)^2$. Notice that by the assumption of $H_G$,

$$\left| W_i - \widetilde{W}_i \right| = \left| H_G\left(\hat{f}^{so}(X_i^{ta}), (Y_i^{ta})\right) - H_G\left(f^{so}(X_i^{ta}), Y_i^{ta}\right) \right| \leq L \left| \hat{f}^{so}(X_i^{ta}) - f^{so}(X_i^{ta}) \right| \tag{6}$$

Plugging (6) into (5), we obtain our desired result. $\qquad\square$

## A.2  Proof of Theorem 2

For simplicity, let $K_h(\cdot) = K(\cdot/h)$ and define the expected regression estimate $\tilde{f} = \sum_{i=1}^n w_i f(X_i)$. To prove Theorem 2, we first give some standard supporting lemmas for kernel smoothing.

**Lemma 1 (Lemma 1 of [Kpotufe and Garg, 2013])** *Under the same assumptions in Theorem 2, for all $x$ with $||x||_2 \leq \triangle_X$, if $f$ is $(\lambda, \alpha)$ Hölder , then, for any $h > 0$, we have $|\tilde{f}(x) - f(x)|^2 \leq \lambda^2 h^{2\alpha}$.*

**Lemma 2 (Corollary of Lemma 3 and Lemma 7 of [Kpotufe and Garg, 2013])** *Under the same assumptions in Theorem 2, let $0 < \delta < 1/6$, for all $x : ||x||_2 \leq \triangle_X$ and $h > 0$, with probability at least $1 - \delta$, we have*

$$|\hat{f}(x) - \tilde{f}(x)|^2 = O\left( \frac{\log(1/\delta)}{nh^d} \right).$$

*Proof of Theorem 2.* we prove Theorem 2 by bounding each corresponding term in Theorem 1. First, by Lemma 1 and Lemma 2, we have for all $x$, with probability at least $1 - \delta$

$$\left| \hat{f}^{so}(x) - f^{so}(x) \right|^2 = O\left( h_{so}^{2\alpha_{so}} + \frac{\log(1/\delta)}{n_{so}h_{so}^d} \right).$$

Specifically, for $X_1^{ta}, \ldots, X_{n_{ta}}^{ta}$, we have

$$\max_{i=1,\cdots,n_{ta}} \left| \hat{f}^{so}(X_i^{ta}) - f^{so}(X_i^{ta}) \right|^2 = O\left( h_{so}^{2\alpha_{so}} + \frac{\log(1/\delta)}{n_{so}h_{so}^d} \right). \tag{7}$$

Next, according to Assumption 1 and 2, $H_G$ is bounded and unbiased and $w_G$ is bounded, we can view $\left\{ \left(X_i^{ta}, \widetilde{W}_i\right) \right\}_{i=1}^{n_{ta}}$ a training set for function $w_G$ that $\widetilde{W}_i = w_G(X_i^{ta}) + \epsilon_{w_G}$ where $\mathbf{E}\left[\epsilon_{w_G}\right] = 0$ and $|\epsilon_{w_G}| \leq 2B$. Based on this observation, using Lemma 1 and Lemma 2 again, for all $x : ||x||_2 \leq \triangle_X$, we have with probability at least $1 - \delta$

$$|\tilde{w}_G(x) - w_G(x)|^2 = O\left( h_{w_G}^{2\alpha_{w_G}} + \frac{\log(1/\delta)}{n_{ta}h_{w_G}^d} \right).$$

Now we are left bounding $\left\|\mathcal{A}_{w_G}(\mathcal{T}) - \mathcal{A}_{w_G}\left(\widetilde{\mathcal{T}}\right)\right\|_\infty$. Notice that for $\mathcal{T}, \widetilde{\mathcal{T}}$ in Theorem 1, and for all $x : \|x\|_2 \le \triangle_X$:

$$\left|\mathcal{A}_{w_G}(\mathcal{T})(x) - \mathcal{A}_{w_G}\left(\widetilde{\mathcal{T}}\right)(x)\right| = \frac{\sum_{i=1}^{n_{ta}} K_h\left(\|x - X_i^{ta}\|_2\right)\left(W_i - \widetilde{W}_i\right)}{\sum_{i=1}^{n_{ta}} K_h\left(\|x - X_i^{ta}\|_2\right)}$$

$$= \frac{\sum_{i=1}^{n_{ta}} K_h\left(\|x - X_i^{ta}\|_2\right)\left|W_i - \widetilde{W}_i\right|}{\sum_{i=1}^{n_{ta}} K_h\left(\|x - X_i^{ta}\|_2\right)}$$

$$\triangleq \sum_{i=1}^{n_{ta}} c_i \left|W_i - \widetilde{W}_i\right|$$

for $c_i = \frac{K_h\left(\|x - X_i^{ta}\|_2\right)}{\sum_{i=1}^{n_{ta}} K_h\left(\|x - X_i^{ta}\|_2\right)}$. Now according to Theorem 1, we only need to bound $\left(\sum_{i=1}^{n_{ta}} c_i \left|\hat{f}^{so}\left(X_i^{ta}\right) - f^{so}\left(X_i^{ta}\right)\right|\right)^2$. With probability at least $1 - \delta$, we have:

$$\left(\sum_{i=1}^{n_{ta}} c_i \left|\hat{f}^{so}\left(X_i^{ta}\right) - f^{so}\left(X_i^{ta}\right)\right|\right)^2 \le \left(\sum_{i=1}^{n_{ta}} c_i\right)^2 \left(\max_{i=1,\cdots,n_{ta}} \left|\hat{f}^{so}\left(X_i^{ta}\right) - f^{so}\left(X_i^{ta}\right)\right|^2\right) \quad (8)$$

$$= \max_{i=1,\cdots,n_{ta}} \left|\hat{f}^{so}\left(X_i^{ta}\right) - f^{so}\left(X_i^{ta}\right)\right|^2 \quad (9)$$

$$= O\left(h_{so}^{2\alpha_{so}} + \frac{\log(n_{ta}/\delta)}{n_{so}h_{so}^d}\right), \quad (10)$$

where (8) is because maximum is bigger than other terms, (9) is because $\sum_{i=1}^{n_{ta}} c_i = 1$ by definition, and (10) is by (7). Putting these all together, using Theorem 1 and choosing the bandwidth according to Theorem 2, we can show for all $x : \|x\|_2 \le \triangle_X$

$$\left|f^{ta}(x) - \hat{f}^{ta}(x)\right|^2 = O\left(n_{so}^{\frac{-2\alpha_{so}}{2\alpha_{so}+d}} + n_{ta}^{\frac{-2\alpha_{w_G}}{2\alpha_{w_G}+d}}\right)\log\left(\frac{1}{\delta}\right).$$

Now integrate with respect to $P_{X^{ta}}$ we obtain our desired result. $\qquad\square$

## A.3 Proof of Theorem 3

The proof strategy is similar to that of Theorem 2. Using Theorem 1 we have

$$\mathbf{E}\left[\left|\hat{f}^{ta}(X) - f^{ta}(X)\right|^2\right] = O\left(\mathbf{E}\left[\left|\hat{f}^{so}(X) - f^{so}(X)\right|^2 + |\widetilde{w}_G(X) - w_G(X)|^2 + \right.\right.$$

$$\left.\left.\left(\sum_{i=1}^{n_{ta}} c_i\left(X_i^{ta}\right)\left|\hat{f}^{so}\left(X_i^{ta}\right) - f^{so}\left(X_i^{ta}\right)\right|\right)^2\right]\right).$$

where the expectation is taken over $P_{x^{ta}}$ and $\mathcal{T}^{ta}$. Now we bound three terms on the right hand side separately. By Corollary 3 of Steinwart et al. [2009], we have with probability at least $1 - \delta$

$$\mathbf{E}\left[\left|\hat{f}^{so}(X) - f^{so}(X)\right|^2\right] = O\left(\lambda_{so}^{\beta_{so}} + \frac{\log(1/\delta)}{\lambda_{so}^p n_{so}}\right), \quad (11)$$

where expectation is taken over $P_x^{ta}$. Taking union bound over $X_1^{ta}, \ldots, X_{n_{ta}}^{ta}$, we have

$$\max_{i=1,\cdots,n_{ta}} \mathbf{E}\left[\left|\hat{f}^{so}(X_i^{ta}) - f^{so}(X_i^{ta})\right|^2\right] = O\left(\lambda_{so}^{\beta_{so}} + \frac{\log(n_{ta}/\delta)}{\lambda_{so}^p n_{so}}\right). \quad (12)$$

where the expectation is taken over $\mathcal{T}^{ta}$. Next, using the exactly same argument as in the Theorem 2, we can view $\left\{\left(X_i^{ta}, \widetilde{W}_i\right)\right\}_{i=1}^{n_{ta}}$ a training set for function $w_G$ that $\widetilde{W}_i = w_G(X_i^{ta}) + \epsilon_{w_G}$ as

$\widetilde{W}_i = w_G\left(X_i^{ta}\right) + \epsilon_{w_G}$ where $\mathbf{E}\left[\epsilon_{w_G}\right] = 0$ and $\left|\epsilon_{w_G}\right| \leq 2B$. Thus applying Corollary 3 of Steinwart et al. [2009] again, we have with probability at least $1 - \delta$

$$\mathbf{E}\left[\left|\tilde{w}_G\left(X\right) - w_G\left(X\right)\right|^2\right] = O\left(\lambda_{w_G}^{\beta_{w_G}} + \frac{\log\left(1/\delta\right)}{\lambda_{w_G}^p n_{ta}}\right).$$

where expectation is taken over $P_{x^{ta}}$. Now we analyze the stability of KRR. We use $\Phi\left(x\right)$ to denotes the feature map corresponding with the given kernel $K$ so $K(x,y) = \Phi\left(x\right)^\top \Phi\left(y\right)$. Also for simplicity, we denote

$$\mathbf{\Phi}_{ta} = \left(\Phi\left(x_1^{ta}\right) \mid \cdots \mid \Phi\left(x_{n_{ta}}^{ta}\right)\right)$$

the feature matrix of target domain data. With these notations, we can write

$$\left|\mathcal{A}_{w_G}\left(\mathcal{T}^{w_G}\right)\left(x\right) - \mathcal{A}_{w_G}\left(\widetilde{\mathcal{T}}^{w_G}\right)\left(x\right)\right|$$

$$= \left|\begin{pmatrix} W_1 - \widetilde{W}_1 \\ \cdots \\ W_{n_{ta}} - \widetilde{W}_{n_{ta}} \end{pmatrix}^\top \left(\mathbf{\Phi}_{ta}^\top \mathbf{\Phi} + n_{ta}\lambda_{w_G}\mathbf{I}\right)^{-1} \mathbf{\Phi}_{ta}^\top \Phi\left(x\right)\right|$$

$$= \left|\left(\mathbf{\Phi}_{ta}\begin{pmatrix} W_1 - \widetilde{W}_1 \\ \cdots \\ W_{n_{ta}} - \widetilde{W}_{n_{ta}} \end{pmatrix}\right)^\top \left(\mathbf{\Phi}_{ta}\mathbf{\Phi}_{ta}^\top + n_{ta}\lambda_{w_G}\mathbf{I}\right)^{-1} \Phi\left(x\right)\right|$$

$$\leq \left\|\begin{pmatrix} k^{1/2}\left|W_1 - \widetilde{W}_1\right| \\ \cdots \\ k^{1/2}\left|W_{n_{ta}} - \widetilde{W}_{n_{ta}}\right| \end{pmatrix}\right\|_2 \left\|\left(\mathbf{\Phi}_{ta}\mathbf{\Phi}_{ta}^\top + n_{ta}\lambda_{w_G}\mathbf{I}\right)^{-1}\right\|_{op} k^{1/2}$$

$$\leq \sum_{i=1}^{n_{ta}} \frac{k}{n_{ta}\lambda_{w_G}}\left|W_i - \widetilde{W}_i\right|$$

$$\triangleq \sum_{i=1}^{n_{ta}} c_i \left|W_i - \widetilde{W}_i\right|.$$

The second equality we used the identity that $\left(\mathbf{\Phi}^\top \mathbf{\Phi} + \lambda\mathbf{I}\right)^{-1}\mathbf{\Phi}^\top = \mathbf{\Phi}^\top\left(\mathbf{\Phi}\mathbf{\Phi}^\top + \lambda\mathbf{I}\right)^{-1}$ for any $\mathbf{\Phi}$ and $\lambda$. The first inequality we used sub-multiplicity of operator norm and the assumption $\left\|\Phi\left(x\right)\right\|_{\mathcal{H}} \leq k^{1/2}$. The second inequality we used the fact the lower bound of least eigenvalue of $\left(\mathbf{\Phi}_{ta}\mathbf{\Phi}_{ta}^\top + n_{ta}\lambda_{w_G}\mathbf{I}\right)$ is $n_{ta}\lambda_{w_G}$. Therefore, applying Cauchy-Schwartz inequality and using the bound in (12), we have with probability at least $1 - \delta$,

$$\mathbf{E}\left[\left(\sum_{i=1}^{n_{ta}} c_i \left|\hat{f}^{so}\left(X_i^{ta}\right) - f^{so}\left(X_i^{ta}\right)\right|\right)^2\right] \leq \left(\sum_{i=1}^{n_{ta}} c_i^2\right) \cdot \left(\sum_{i=1}^{n_{ta}}\left|\hat{f}^{so}\left(X_i^{ta}\right) - f^{so}\left(X_i^{ta}\right)\right|^2\right)$$

$$= \sum_{i=1}^{n_{ta}} \frac{k^2}{n_{ta}^2\lambda_{w_G}^2} \cdot \mathbf{E}\left[\sum_{i=1}^{n_{ta}}\left|\hat{f}^{so}\left(X_i^{ta}\right) - f^{so}\left(X_i^{ta}\right)\right|^2\right]$$

$$\leq \frac{k^2}{\lambda_{w_G}^2} \cdot \max_{i=1,\ldots,n_{ta}} \mathbf{E}\left[\left|\hat{f}^{so}\left(X_i^{ta}\right) - f^{so}\left(X_i^{ta}\right)\right|^2\right]$$

$$= O\left(\frac{k^2}{\lambda_{w_G}^2}\left(\lambda_{so}^{\beta_{so}} + \frac{\log\left(n_{ta}/\delta\right)}{\lambda_{so}^p n_{so}}\right)\right).$$

Now putting these all together and choosing $\lambda_{so}$ and $\lambda_{w_G}$ according to Theorem 3, we obtain the desired result. $\qquad\square$

## A.4 Proof of Theorem 4

We first prove a general theorem for cross-validation. This is a standard result for cross-validation and we include the proof for completeness.

**Theorem 5** *Let $\Theta$ be the set of all hypotheses and $\hat{\theta} = \text{argmin}_{\theta \in \Theta} \sum_{i=1}^{n_{val}} \left( \hat{f}_{\theta}^{ta} \left( X_i^{val} \right) - Y_i^{val} \right)^2$ the estimator that minimizes error on the cross-validation set. Then with probability at least $1 - \delta$:*

$$\mathbf{E}\left[ R\left( \hat{f}_{\hat{\theta}}^{ta} \right) \right] - R\left( f^{ta} \right) = O\left( \mathbf{E}\left[ R\left( \hat{f}_{\theta_{\star}}^{ta} \right) \right] - R\left( f^{ta} \right) + \frac{\log \frac{|\Theta|}{\delta}}{n_{val}} \right),$$

*where $\theta^* = \text{argmin}_{\theta \in \Theta} R\left( \hat{f}_{\theta} \right)$ and the expectation is taken over $\mathcal{T}^{so}$ and $\mathcal{T}^{ta}$.*

To prove of Theorem 5, we use the following type of Bernstein's inequality [Craig, 1933]:

**Lemma 3** *Let $X_1, \ldots, X_n$ be random variables and suppose that for $k \geq 3$:*

$$\mathbf{E}[|X_i - \mathbf{E}[X_i]|^k] \leq \frac{\mathbf{Var}[X_i]}{2} k! r^{k-2},$$

*for some $r > 0$. Then with probability $> 1 - \delta$:*

$$\frac{1}{n} \sum_{i=1}^{n} (X_i - \mathbf{E}[X_i]) \leq \frac{\log(1/\delta)}{nt} + \frac{t\mathbf{Var}[X_i]}{2(1-c)},$$

*for $0 \leq tr \leq c < 1$.*

*Proof of Theorem 5:* For a given $\theta \in \Theta$, we obtain a corresponding estimated regression function $\hat{f}_{\theta}$. Define $U_i^{\theta} \triangleq - \left( Y_i^{val} - \hat{f}_{\theta}^{ta}(X_i^{val}) \right)^2 + (Y_i - f^{ta}(X_i^{val}))^2$. Compute the expectation:

$$\begin{aligned}
\mathbf{E}\left[ U_i^{\theta} \right] &= - \mathbf{E}\left[ -2Y_i^{val} \hat{f}_{\theta}^{ta} \left( X_i^{val} \right) + \hat{f}_{\theta}^{ta} \left( X_i^{val} \right)^2 + 2Y_i^{val} f^{ta} \left( X_i^{val} \right) - f^{ta} \left( X_i^{val} \right)^2 \right] \\
&= - \mathbf{E}\left[ \left( \hat{f}_{\theta}^{ta} \left( X_i^{val} \right) - f^{ta} \left( X_i^{val} \right) \right)^2 \right] \\
&= R\left( f^{ta} \right) - R\left( \hat{f}_{\theta}^{ta} \right).
\end{aligned}$$

Also, by definition, it is easy to see

$$\frac{1}{n_{val}} \sum_{i=1}^{n_{val}} U_i^{\theta} = \hat{R}\left( f \right) - \hat{R}\left( \hat{f}_{\theta}^{ta} \right).$$

In order to apply Bernstein's inequality, we must first bound the variance of $U_i^{\theta}$:

$$\begin{aligned}
\mathbf{var}\left[ U_i^{\theta} \right] &\leq \mathbf{E}\left[ (U_i^{\theta})^2 \right] \\
&= \mathbf{E}\left[ \left( - \left( Y_i^{val} - \hat{f}_{\theta}^{val} \left( X_i^{val} \right) \right)^2 + \left( Y_i^{val} - f^{ta} \left( X_i^{val} \right) \right)^2 \right)^2 \right] \\
&= \mathbf{E}\left[ \left( f^{ta} \left( X_i \right) - \hat{f}_{\theta}^{ta} \right)^4 + 4\epsilon_i \left( f^{ta} \left( X_i^{vak} \right) - \hat{f}_{\theta}^{ta} \left( X_i^{val} \right) \right)^3 + 4\epsilon_i^2 \left( f^{ta} \left( X_i^{val} \right) - \hat{f}_{\theta}^{ta} \left( X_i^{val} \right) \right)^2 \right] \\
&\leq -4\triangle_Y^2 \mathbf{E}\left[ U_i \right]
\end{aligned}$$

where in the last inequality we used the domain of $Y$ is bounded. Since $\mathbf{U}_i$ is a sum of bounded random variables, the moment condition is satisfied with $r = 4\triangle_Y^2$. Now apply Craig-Bernstein inequality to $U_i^{\theta}$s, with probability at least $1 - \delta$:

$$\frac{1}{n_{val}} \sum_{i=1}^{n_{val}} U_i^{\theta} - \mathbf{E}\left[ U_i^{\theta} \right] \leq \frac{\log(1/\delta)}{n_{val}t} + \frac{-2t\triangle_Y^2 \mathbf{E}\left[ U_i^{\theta} \right]}{1-c}.$$

We need to ensure that $c < 1$. To do this, let $c = tr = 4t\triangle_Y^2$ and let $t < \frac{1}{6\triangle_Y^2}$, then it is easy to see that $c < 1$. For simplicity, define $a = \frac{2t\triangle_Y^2}{1-c} < 1$. Now grouping terms we get:

$$(1-a)\left(-\mathbf{E}\left[U_i^\theta\right]\right) + \frac{1}{n_{val}} \sum_{i=1}^{n_{val}} U_i^\theta \leq \frac{\log(1/\delta)}{n_{val}t}$$

$$(1-a)\left(R\left(\hat{f}^{ta}\right) - R(f)\right) - \left(\hat{R}\left(f_\theta^{ta}\right) - \hat{R}(f)\right) \leq \frac{\log(1/\delta)}{n_{val}t}$$

$$R\left(\hat{f}_\theta^{ta}\right) - R\left(f^{ta}\right) \leq \frac{1}{1-a}\left(\hat{R}\left(\hat{f}_\theta\right) - \hat{R}\left(f^{ta}\right) + \frac{\log(1/\delta)}{n_{val}t}\right).$$

Take union bound over $\Theta$, and consider $\hat{f}_{\hat{\theta}}$:

$$R\left(\hat{f}_{\hat{\theta}}^{ta}\right) - R\left(f^{ta}\right) \leq \frac{1}{1-a}\left(\hat{R}\left(\hat{f}_{\hat{\theta}}^{ta}\right) - \hat{R}\left(f^{ta}\right) + \frac{\log(|\Theta|/\delta)}{n_{val}t}\right).$$

Now, recall that $\hat{f}_{\hat{\theta}}^{ta}$ is the minimizer for $\hat{R}$ among all estimators induced by $\Theta$, we have

$$R\left(\hat{f}_{\hat{\theta}}^{ta}\right) - R\left(f^{ta}\right) \leq \frac{1}{1-a}\left(\hat{R}\left(\hat{f}_{\theta_\star}^{ta}\right) - \hat{R}\left(f^{ta}\right) + \frac{\log(|\Theta|/\delta)}{n_{val}t}\right).$$

Now taking expectation over $\mathcal{T}^{val}$ then over $\mathcal{T}^{so}$ and $\mathcal{T}^{ta}$ we obtain the desired result. $\qquad \square$

Now we are ready to prove Theorem 4. Since $\overline{\mathcal{G}}$ is an $\epsilon$-cover of $\mathcal{G}$, there exists $G' \in \overline{\mathcal{G}}$ such that $||G' - G^\star||_\infty \leq \epsilon$. For any $x$,

$$\left|f^{ta}(x) - \hat{f}_{G'}^{ta}(x)\right|$$

$$= \left|G^\star\left(f^{so}(x), w_{G^\star}(x)\right) - G'\left(\hat{f}^{so}(x), \hat{w}_{G'}(x)\right)\right|$$

$$\leq \left|G^\star\left(f^{so}(x), w_{G^\star}(x)\right) - G^\star\left(\hat{f}^{so}(x), \hat{w}_{G^\star}(x)\right)\right| + \left|G^\star\left(\hat{f}^{so}(x), \hat{w}_{G^\star}(x)\right) - G'\left(\hat{f}^{so}(x), \hat{w}_{G^\star}(x)\right)\right|$$

$$+ \left|G'\left(\hat{f}^{so}(x), \hat{w}_{G^\star}(x)\right) - G'\left(\hat{f}^{so}(x), \hat{w}_{G'}(x)\right)\right| \tag{13}$$

where $\hat{w}_{G^\star} = \mathcal{A}_{w_G}\left(\{X_i^{ta}, W_i^\star\}\right)$ and $W_i^\star = H_{G'}\left(\hat{f}^{so}(X_i^\star), Y_i^\star\right) + w_{G^\star}(X_i^{ta}) - w_{G'}(X_i^{ta})$, i.e. an un-biased estimated of $w_{G^\star}(X_i^\star)$. We can bound three terms in (13) separately. The first term is just the difference between estimator based on $G^\star$ and the true $f^{ta}$, so after taking expectation it becomes the excess risk of $\hat{f}_{G^\star}^{ta}$. By our construction of $\epsilon$-cover of $\mathcal{G}$, the second term is smaller than $\epsilon$. For the third term, notice that by Lipschitz assumption on $G$s and our assumptions on $G$s in $\mathcal{G}$ in the theorem 4, we have:

$$\left|G'\left(\hat{f}^{so}(x), \hat{w}_{G^\star}(x)\right) - G'\left(\hat{f}^{so}(x), \hat{w}_{G'}(x)\right)\right|$$

$$\leq L\left(|\hat{w}_{G^\star}(x) - \hat{w}_{G'}(x)|\right)$$

$$\leq L^2 \sum_{i=1}^{n_{ta}} c_i \, ||G^\star - G'||_\infty$$

$$= O\left(\sum_{i=1}^{n_{ta}} c_i \epsilon\right).$$

Now we have shown $R\left(\hat{f}_{G'}^{ta}\right) - R(f^{ta}) = O\left(R\left(\hat{f}_{G^\star}^{ta}\right) - R(f^{ta})\right)$. Let $\overline{G}_\star = \operatorname{argmin}_{G \in \overline{\mathcal{G}}} R\left(\hat{f}_G\right)$, the best transformation function in $\overline{\mathcal{G}}$. By the optimality of $\overline{G}_\star$, we have $R\left(\hat{f}_{\overline{G}_\star}^{ta}\right) - R(f^{ta}) = O\left(R\left(\hat{f}_{G^\star}^{ta}\right) - R(f^{ta})\right)$. Applying Theorem 5 with our assumptions on $\epsilon$ and $n_{val}$ we know $R\left(\hat{f}_{\overline{G}^\star}^{ta}\right) - R(f^{ta}) = O\left(R\left(\hat{f}_{\overline{G}_\star}^{ta}\right) - R(f^{ta})\right)$. Combing these facts we have $R\left(\hat{f}_{\overline{G}^\star}^{ta}\right) - R(f^{ta}) = O\left(R\left(\hat{f}_{G_\star}^{ta}\right) - R(f^{ta})\right)$.

# B  Regression Calibration for Measurement Error Problem

Given, $f^{so}$, in this section we provide a standard technique to obtain an unbiased estimate of $w_G\left(X_i^{ta}\right)$s. Since we assume

$$Y^{ta} = f^{ta}\left(X^{ta}\right) + \epsilon^{ta},$$

the measurement error model corresponds to *classical error model* in Carroll et al. [2006]. Regression calibration is a widely used and reasonably well investigated method for measurement error problem. The algorithm is as follows (we have adapted the general algorithm to our HTL problem):

- Compute an estimate of $f^{ta}\left(X_i^{ta}\right)$: $\tilde{f}^{ta}\left(X_i^{ta}\right)$. Note that directly using $Y_i^{ta}$ is one of the option for $\tilde{f}^{ta}\left(X_i^{ta}\right)$.

- Compute $G_{f^{so}\left(X_i^{ta}\right)}^{-1}\left(\tilde{f}^{ta}\left(X_i^{ta}\right)\right)$.

- Calibrate our previous computed value by applying some function $F$:

$$\widetilde{W}_i = F\left(G_{f^{so}\left(X_i^{ta}\right)}^{-1}\left(\tilde{f}^{ta}\left(X_i^{ta}\right)\right)\right)$$

where $F$ depends on $G$ and the specific distribution on noise.

Now we consider the loglinear mean model as a concrete example. Suppose

$$G\left(f^{so}\left(x\right), w_G\left(x\right)\right) = \beta f^{so}\left(x\right)\log\left(w_G\left(x\right)\right)$$

where $\beta$ is some constant. Further, we assume $\epsilon^{ta} \sim \mathcal{N}\left(0, \sigma^2\right)$ Now we apply the regression calibration algorithm.

- First we choose $Y_i^{ta}$ as our estimate for $\tilde{f}^{ta}\left(X_i^{ta}\right)$.

- Second, by our choice of $G$:

$$G_{f^{so}\left(X_i^{ta}\right)}^{-1}\left(Y_i^{ta}\right) = \exp\left(\frac{Y_i^{ta}}{\beta f^{so}\left(X_i^{ta}\right)}\right)$$

- Last, for our choice of $G$ and assumption of $\epsilon^{ta}$, the corresponding $F$ and final estimate of $w_G\left(X_i^{ta}\right)$ is

$$\begin{aligned}
\widetilde{W}_i &= F\left(G_{f^{so}\left(X_i^{ta}\right)}^{-1}\left(\tilde{f}^{ta}\left(X_i^{ta}\right)\right)\right) \\
&= \exp\left(\log\left(G_{f^{so}\left(X_i^{ta}\right)}^{-1}\left(\tilde{f}^{ta}\left(X_i^{ta}\right)\right)\right) + \sigma^2\left(f^{so}\left(X_i^{ta}\right)\right)^2\right) \\
&= \exp\left(\frac{Y_i^{ta}}{\beta f^{so}\left(X_i^{ta}\right)} + \sigma^2\left(f^{so}\left(X_i^{ta}\right)\right)^2\right).
\end{aligned}$$

The estimator for $w_G\left(X_i^{ta}\right)$ depends on some distribution specific parameters which may be unknown, like $\sigma^2$ in the previous example. In such cases, we may replace these parameters by our estimates. For example, in the previous Gaussian noise case, suppose for each $X_i^{ta}$, we have multiple observations $\{Y_{ij}\}_{j=1}^{n_i}$. Then we can estimate $\sigma^2$ by

$$\hat{\sigma}^2 = \frac{\sum_{i=1}^{n_{ta}}\sum_{j=1}^{n_i}\left(Y_{ji}^{ta} - \bar{Y}_i^{ta}\right)^2}{\sum_{i=1}^{n_{ta}}\left(n_i - 1\right)}$$

where $\bar{Y}_i^{ta} = \frac{\sum_{j=1}^{n_i} Y_{ij}}{n_i}$.

Here we only provide one method for measurement error problem. There are other techniques such as simulation extrapolation and likelihood method which may be also applicable in many situations. The choice of method depends on specific transformation $G$ and assumptions on the distribution of the noise. Again, interested readers are referred to Carroll et al. [2006] for details.

| | $n_{ta} = 10$ | $n_{ta} = 20$ | $n_{ta} = 40$ | $n_{ta} = 80$ | $n_{ta} = 160$ | $n_{ta} = 320$ |
|---|---|---|---|---|---|---|
| Only Target KS | $0.005 \pm 0.001$ | $0.003 \pm 0.001$ | $0.003 \pm 0.001$ | $0.003 \pm 0.000$ | $0.002 \pm 0.000$ | $0.002 \pm 0.000$ |
| Only Target KRR | $\mathbf{0.001 \pm 0.001}$ | $\mathbf{0.001 \pm 0.000}$ | $\mathbf{0.000 \pm 0.000}$ | $\mathbf{0.000 \pm 0.000}$ | $\mathbf{0.000 \pm 0.000}$ | $\mathbf{0.000 \pm 0.000}$ |
| Only Source KS | $0.031 \pm 0.012$ | $0.031 \pm 0.012$ | $0.031 \pm 0.012$ | $0.031 \pm 0.012$ | $0.031 \pm 0.012$ | $0.031 \pm 0.012$ |
| Only Source KRR | $0.016 \pm 0.013$ | $0.016 \pm 0.013$ | $0.016 \pm 0.013$ | $0.016 \pm 0.013$ | $0.016 \pm 0.013$ | $0.016 \pm 0.013$ |
| Combined KS | $0.023 \pm 0.017$ | $0.029 \pm 0.011$ | $0.017 \pm 0.013$ | $0.007 \pm 0.007$ | $0.002 \pm 0.000$ | $0.002 \pm 0.000$ |
| Combined KRR | $0.006 \pm 0.008$ | $0.009 \pm 0.010$ | $0.002 \pm 0.002$ | $0.001 \pm 0.000$ | $0.001 \pm 0.000$ | $0.001 \pm 0.000$ |
| CDM | $0.004 \pm 0.002$ | $0.007 \pm 0.001$ | $0.004 \pm 0.002$ | $0.001 \pm 0.000$ | $0.001 \pm 0.000$ | $0.012 \pm 0.002$ |
| Offset KS | $0.003 \pm 0.001$ | $0.002 \pm 0.001$ | $0.002 \pm 0.000$ | $0.002 \pm 0.000$ | $0.002 \pm 0.000$ | $0.001 \pm 0.000$ |
| Offset KRR | $0.002 \pm 0.001$ | $\mathbf{0.001 \pm 0.000}$ | $\mathbf{0.000 \pm 0.000}$ | $\mathbf{0.000 \pm 0.000}$ | $\mathbf{0.000 \pm 0.000}$ | $\mathbf{0.000 \pm 0.000}$ |
| Scale KS | $0.004 \pm 0.002$ | $0.003 \pm 0.001$ | $0.002 \pm 0.001$ | $0.002 \pm 0.000$ | $0.002 \pm 0.000$ | $0.002 \pm 0.000$ |
| Scale KRR | $\mathbf{0.001 \pm 0.000}$ | $\mathbf{0.001 \pm 0.000}$ | $\mathbf{0.000 \pm 0.000}$ | $\mathbf{0.000 \pm 0.000}$ | $\mathbf{0.000 \pm 0.000}$ | $\mathbf{0.000 \pm 0.000}$ |

Table 3: 1 standard deviation intervals for the mean squared errors of various algorithms when transferring from kin-8nh to kin-8fm. The values in bold are the best errors for each $n_{ta}$.

## C    Additional Experimental Results

### C.1    Synthetic data

This section gives details of the synthetic data. For both experiments, we use $n_{so} = 10000$ samples from the source domain, and $n_{ta} = 100$ samples from the target domain. We put Gaussian noise on the labels: $\epsilon^{so} \sim \mathcal{N}(0, 0.01)$, $\epsilon^{ta} \sim \mathcal{N}(0, 0.01)$; and we use KS with a gaussian kernel for estimating $f^{so}$ and $w_G$.

Figure 1b shows the offset example in Section 3, where we consider

$$f^{so}(x) = \sqrt{x(1-x)} \sin\left(\frac{2.1\pi}{x + 0.05}\right), f^{ta}(x) = f^{so}(x) + x.$$

We used the transformation function $G(a, b) = a + b$. The bandwidths of the kernels were chosen by cross validation. For estimating $f^{so}$, the chosen bandwidth is $h_{so} = 10^{-8}$, and for estimating $w_G$, the chosen value is $h_{w_G} = 10^{-5}$. Figure 1c shows the scale example in Section 3, where we consider the same source regression function and $f^{ta}(x) = 5 f^{so}(x)$. We tested the transformation function $G(a, b) = ab$. Bandwidth parameters were again chosen by cross validation: $h_{so} = 10^{-7}$ for estimating $f^{so}$, and $h_{w_G} = 5 \times 10^{-4}$ for estimating $w_G$. The plots show that by using our proposed transfer learning framework with an appropriate transformation function, we can estimate the target regression function better, especially in regions where $f^{ta}$ is not smooth.

### C.2    Transferring from kin-8nh to kin-8fm

Now we briefly discuss the results of the second transfer task with the robotic arm data described in Section 6. The source domain is kin-8nh and the target domain is kin-8fm. The results are shown in Table 3. Here we see the effects of trying to transfer to an "easy" domain. We do not gain any advantage by using the transfer algorithm, except for the smallest value of $n_{ta}$ - even here the gain is minimal. However, it should be noted that using transfer learning does not negatively affect performance. And we point out that in a dataset where the smoothness conditions are unknown, we would use cross-validation to decide whether or not to use the source data.