[Reviews · NeurIPS 2017]

Reviewer 1



The papers tackles the issue of domain adaptation regression some target labels are available. They focus on non-parametric regression when the source and target regression functions are different. Basically, they assume there is a transformation function between the source and target regression functions. Given this transformation, the algorithm proposed works as follows. 1. train the source domain regression function (with an algo) 2 construct new "target" data thanks to the transformation function G(.,.), the source function and the target sample 3 train an auxiliary function with this new dataset (with another algo) 4 the final regression target model returned by the algorithm is defined as G(source model(x),auxiliary function(x)) They also provide a strong theoretical study and experiments. PROS: - The theoretical results (based on stability) seem solid - The experiments provided are sound CONS: - My main concern is related to the fact that the method depends on the transformation function G which can be a disadvantage of the algorithm in practice. (However it is not hidden by the authors and they discuss the definition of such a function.)

Reviewer 2



The paper presents a supervised non-parametric hypothesis transfer learning (HTL) approach for regression and its analysis, aimed at the cases where one has plenty of training data coming from the source task and few examples from the target one. The paper makes an assumption that the source and the target regression functions are related through so called transformation function (TF). The TF is assumed to have some parametric form (e.g. linking regressions functions linearly) and the goal of an algorithm is to recover its parameters. Once these parameters are learned, the hypothesis trained on the source task can be transformed to the hypothesis designated for the target task. The paper proposes two ways for estimation of these parameters, that is through kernel smoothing and kernel ridge regression. For both cases the paper presents consistency bounds, showing that one can have an improvement in the exponent of the non-parametric rate once the TF is smoother, in Holder sense, than the target regression function. In addition, the paper argues that one can run model selection over the class of TF through cross-validation and shows some bounds to justify this, which is not particularly surprising. Finally, the paper concludes with numerical experiments on two regression tasks on the real-world small-scale datasets. The rates stated in main theorems, 2 and 3 appear to make sense. One can see this from theorem 1, which is used to prove both, and essentially states an excess risk as a decomposition into the sum of excess risk bounds for source/TF regression functions. The later are bounded through well established excess risk bounds from the literature. I do not think this is a breakthrough paper, but it has a novel point as it addresses the theory of HTL in the non-parametric setting because it studies the Bayes risk rates rather than generalization bounds as was done previously. The paper has a novel bit as it studies non-parametric consistency rates of HTL, while previous theoretical works focused on the risk bounds. As one should expect, the improvement then occurs in the right place, that is in the exponent of the rate, rather than in some additive or multiplicative term. At the same time the algorithm appears to be simple and easy to implement. This makes this paper a good contribution to the literature on HTL. Also, the paper is quite well positioned in the related work. Unfortunately, it doesn't compare to previous HTL approaches experimentally and experimental evaluation is, frankly, quite modest, despite the simplicity of the algorithm.

Reviewer 3



This paper considers the hypothesis transfer learning problem, which tries to incorporate a hypothesis trained on the source domain into the learning procedure of the target domain. A unified framework for HTL is proposed by introducing the transformation function, and a theoretical study on the excess risk is given. Moreover, two case studies on kernel smoothing and kernel ridge regression show the faster convergence rates of excess risk. The paper is well written, the unified framework and the theoretical studies are novel in this field. My comments are listed as follows: 1) Most examples are simple linear transformation functions. What about nonlinear cases? 2) In the proposed algorithm, training the auxiliary function from T^{WG} is involved. However, when the number of labeled data in target data is small, this may fail to obtain a good auxiliary function. 3) In Section 5, finding the best transformation function with a specific type is discussed. In practice, it would be also desirable to know how to select the type of transformation function. 4) In the experiments, Figure 2 (a) is redundant, since the results are provided in Table 1, I think the space should be saved to present more details of results on fMRI dataset.